# SizeShiftReg: a Regularization Method for Improving Size-Generalization in Graph Neural Networks

**Davide Buffelli**
Department of Information Engineering
University of Padova
Padova, Italy 35131
davide.buffelli@unipd.it

**Pietro Liò**
Department of Computer Science and Technology
University of Cambridge
Cambridge, United Kingdom CB3 0FD
pietro.lio@cl.cam.ac.uk

**Fabio Vandin**
Department of Information Engineering
University of Padova
Padova, Italy 35131
fabio.vandin@unipd.it

## Abstract

In the past few years, graph neural networks (GNNs) have become the *de facto* model of choice for graph classification. While, from the theoretical viewpoint, most GNNs can operate on graphs of any size, it is empirically observed that their classification performance degrades when they are applied on graphs with sizes that differ from those in the training data. Previous works have tried to tackle this issue in graph classification by providing the model with inductive biases derived from assumptions on the generative process of the graphs, or by requiring access to graphs from the test domain. The first strategy is tied to the quality of the assumptions made for the generative process, and requires the use of specific models designed after the explicit definition of the generative process of the data, leaving open the question of how to improve the performance of *generic* GNN models in general settings. On the other hand, the second strategy can be applied to any GNN, but requires access to information that is not always easy to obtain. In this work we consider the scenario in which we only have access to the training data, and we propose a regularization strategy that can be applied to any GNN to improve its generalization capabilities from smaller to larger graphs without requiring access to the test data. Our regularization is based on the idea of simulating a shift in the size of the training graphs using coarsening techniques, and enforcing the model to be robust to such a shift. Experimental results on standard datasets show that popular GNN models, trained on the 50% smallest graphs in the dataset and tested on the 10% largest graphs, obtain performance improvements of up to 30% when trained with our regularization strategy.

## 1   Introduction

Graph structured data are found in a large variety of domains, ranging from social networks, to molecules and transportation networks. When dealing with graph data in machine learning settings, it is common to resort to graph representation learning [29]. Graph representation learning is the task of learning to generate vector representations of the nodes, or of entire graphs, that encode structural and feature-related information that can be used for downstream tasks. In this area, Graph Neural

36th Conference on Neural Information Processing Systems (NeurIPS 2022).

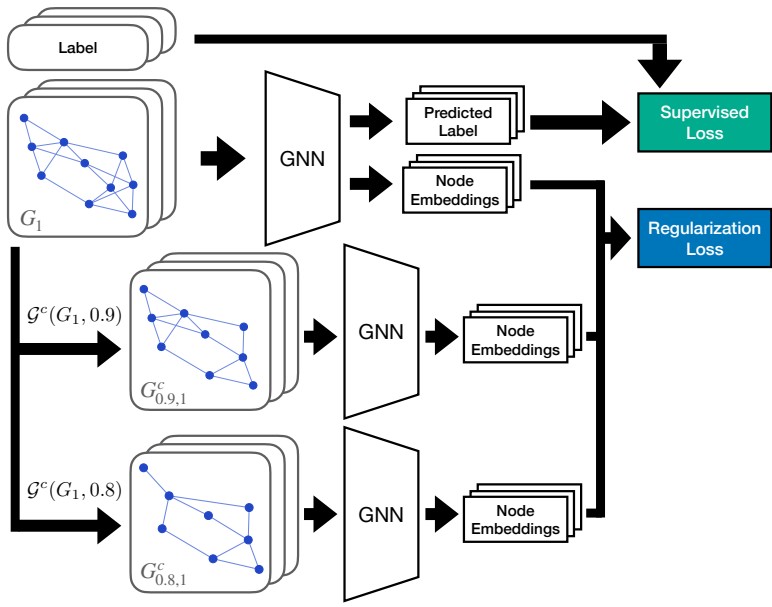

Figure 1: Overview of our method: given a training set of labelled graphs, we simulate a size shift by applying a coarsening function $\mathcal{G}^c$ and we regularize the model to be robust towards this shift. The regularization enforces the *distribution* of node embeddings generated by the model for the original graph and its coarsened versions to be similar.

Networks (GNNs) are a deep learning model for graph representation learning that has become the *de facto* standard for many tasks, including graph classification, which is the focus of this work.

GNNs are designed to be able to work on graphs of any size. However, empirical results show that they struggle at generalizing to graphs of sizes that differ from those in the training data [33, 25, 6, 68]. Obtaining good size-generalization performance from smaller to larger graphs is crucial for several reasons. **First**, it is common for graphs in the same domain to vary in size. For example, social networks can range from tens to millions of nodes, and molecular graphs can go from few atoms, to large compounds with thousands of atoms. **Second**, in many scenarios, obtaining labels for smaller graphs is cheaper than for larger graphs. For example, recently deep learning is being used in combinatorial optimization problems [5], and GNNs are heavily used in this area as many problems can be formulated as graph classification instances [53, 24, 11]. Combinatorial optimization often deals with NP-hard problems, and obtaining ground truth labels can be extremely expensive for large scale graphs. **Third**, training on smaller graphs requires less computational resources.

Two approaches have been proposed in literature to tackle the poor size-generalization of GNNs. The first approach makes assumptions on the causal graph describing the generative process behind the graphs in the dataset, and then designs a model that is tailored towards such causal graph [6]. This strategy requires an explicit definition of the generative process behind the data, which, in the case of Bevilacqua et al. [6], leads to a model requiring the computation of induced homomorphism densities over all connected $k$-vertex subgraphs in order to make a prediction. While this strategy shows great results on synthetic graphs produced by the assumed generative process, its benefits are reduced on real-world graphs, for which the underlying generative process is unknown. The second approach assumes access to graphs coming from different domains, or from the test distribution, and applies domain adaptation techniques to transfer knowledge across domains [68, 17]. These methods do not require ad-hoc models, but require knowledge about each graph's domain, or access to the test distribution, both of which are not always available or easily obtainable in advance.

In this work we consider the graph classification scenario in which we only have access to the training data (as in the first approach described above). However, in contrast to prior work, we do not try to improve the size-generalization capabilities (from smaller to larger graphs) of GNNs by handcrafting models on the base of specific knowledge or assumptions, but we study a general form of regularization that can be applied to any GNN, and that aims at letting the model *learn* to

be robust to size-shifts. The idea behind our regularization is to simulate a shift in the size of the training graphs, and to regularize the model to be robust towards this shift. The size-shift simulation is obtained using graph coarsening techniques, while the regularization is performed by minimizing the discrepancy between the distributions of the node embeddings generated by the GNN model for the original training graphs and their coarsened versions (an overview is shown in Figure 1).

In the experimental section we evaluate GNN models on popular benchmark datasets using a particular train/test split designed to test for size-generalizaiton. In particular, following [68, 6], we train on the 50% smallest graphs and test on the 10% largest. This split leads to an average size of the test graphs which is 3 to 9 times larger than the average size of the training graphs. Our results show that our regularization method applied to standard GNN models (GCN [36], PNA [15], GIN [65]) leads to an average performance improvement on the test set of up to 30% across datasets. Furthermore, these performance improvements show that standard GNN models trained with our regularization can achieve better size-generalization performance than more expensive models. The latter more expensive models were designed after making explicit assumptions on the generative process behind the graphs in the dataset [6], while our method relies on *learning* to be robust towards size-shifts.

**Our contributions.** Our contributions are threefold. (1) We propose a regularization strategy that can be applied to any GNN to improve its size-generalization capabilities on the task of graph classification. (2) We analyze the impact that our strategy has on the embeddings generated by a GNN. (3) We test our strategy on popular GNN models, and compare against previously proposed methods, showing that "standard" GNN models augmented with our regularization strategy achieve comparable or better size-generalization performance than more complex models designed after explicit assumptions on the generative process behind the data.

## 2 Preliminaries

**Notation.** Let $G = (V, E)$ be a graph, where $V$ is the set of nodes with size $n$, and $E \subseteq V \times V$ is the set of edges. For a node $v \in V$, we use $\mathcal{N}_v$ to indicate the set of its one-hop neighbours (i.e., the nodes connected to $v$ by an edge). We represent an *attributed* graph $G$ with a tuple $(\mathbf{A}, \mathbf{X})$, where $\mathbf{A} \in \{0, 1\}^{n \times n}$ is the adjacency matrix (i.e., the element in position $i, j$ is equal to 1 if there is an edge between nodes $i$ and $j$, and zero otherwise), and $\mathbf{X} \in \mathbb{R}^{n \times d}$ is the feature matrix, where the $i$-th row, $\mathbf{X}_i$, contains the $d$-dimensional feature vector for node $i$.

### 2.1 Graph Neural Networks

Most popular Graph Neural Networks (GNNs) [12, 64] models follow the *message-passing* paradigm [26]. Let $\mathbf{H}^{(\ell)} \in \mathbb{R}^{n \times d'}$ be the matrix containing the node representations at layer $\ell$ (i.e., the representation for node $i$ is contained in the $i$-th row $\mathbf{H}_i^{(\ell)}$), with $\mathbf{H}^{(0)} = \mathbf{X}$. Given an aggregation function $\Phi$, which is permutation invariant, and a learnable update function $\Psi$ (usually a neural network), a message passing layer updates the representation of every node $v$ as follows:

$$\mathbf{H}_v^{(\ell+1)} = \Psi(\mathbf{H}_v^{(\ell)}, m_v^{(\ell)}), \text{ with } m_v^{(\ell)} = \Phi(\{\mathbf{H}_u^{(\ell)} \, \forall u \in \mathcal{N}_v\}) \tag{1}$$

where $m_v$ represents the aggregated message received by node $v$ from its neighbours. After $L$ message-passing layers, the final node embeddings $\mathbf{H}^{(L)}$ are used to perform a given task (e.g., they are the input to a network performing the task), and the whole system is trained end-to-end.

### 2.2 Graph Coarsening

Given a graph $G = (V, E)$, a coarsened version $G^c = (V^c, E^c)$ is obtained by partitioning $V$ and grouping together the nodes that belong to the same subset of the partition. More formally, given a partition $P = \{S_1, S_2, \ldots, S_{n\prime}\}$ where $S_i \subseteq V$, $S_i \cap S_j = \emptyset$, $\bigcup_i S_i = V$, we associate a node $s_i$ to each subset $S_i$ and let $V^c = \{s_1, s_2, \ldots, s_{n\prime}\}$ and $E^c = \{(s_i, s_j) \mid$ there is a node in $S_i$ connected to at least one node in $S_j\}$. The goal of graph coarsening is to generate a coarsened version of a graph (i.e., $n\prime < n$) such that specific properties are preserved. For instance, Loukas and Vandergheynst [45] aim at preserving the principal eigenvalues and eigenspaces, while Jin et al. [32] aim at minimizing a distance function that measures the changes in the structure and connectivity that occur in the coarsening process. Jin et al. [32] further prove that their algorithm bounds the spectral distance [35] between the original and a "lifted" version of the coarsened graph.

## 2.3 Central Moment Discrepancy

The Central Moment Discrepancy (CMD) [70] is a metric used to measure the discrepancy between the distribution of high-dimensional random variables. Let $p$ and $q$ be two probability distributions with support in the interval $[a, b]^d$ and let $c_k$ be the $k$-th order moment, then the CMD between $p$ and $q$ is defined as:

$$CMD(p, q) = \frac{1}{|b - a|} \|\mathbb{E}(p) - \mathbb{E}(q)\|_2 + \sum_{k=2}^{\infty} \frac{1}{|b - a|^k} \|c_k(p) - c_k(q)\|_2. \tag{2}$$

At a practical level, we follow Zhu et al. [72], and limit the number of considered moments to 5. To align with the definition of CMD it is possible to treat node embeddings as realizations of *bounded* high dimensional distributions by applying bounded activation functions like tanh or sigmoid at the embeddings layer (as done in [70, 72]).

## 3 Our Method

The high level idea behind our method is to "simulate" a size-shift in the training dataset, and to regularize the GNN to be robust to this shift (an overview is shown in Figure 1). While previous works rely on models that incorporate domain-specific inductive biases and/or are tied to assumptions on the generative process of the data, our method aims at *learning* to be robust towards size-shifts.

**Simulating the shift.** Let $\mathcal{G}^c : (G, r) \to G_r^c$ be a *coarsening function*, that takes as input a graph $G$ with $n$ nodes and a ratio $r \in (0, 1)$, and returns a coarsened version of the graph, $G_r^c$, which has $\lfloor r \times n \rfloor$ nodes. Our method is not bound to a specific coarsening function $\mathcal{G}^c$, any existing method can be used, e.g. [45, 43, 32]. Our method then proceeds in the following way: given a dataset of graphs $\mathcal{D} = \{G_1, G_2, \ldots, G_\ell\}$, a coarsening function $\mathcal{G}^c$, and a set of $k$ coarsening ratios $C = \{r_1, r_2, \ldots, r_k\}$, we create a coarsened dataset $\mathcal{D}_{r_j}$ for each ratio by applying the coarsening function to each graph in the dataset: $\mathcal{D}_{r_j} = \{G_{r_j,1}^c = \mathcal{G}^c(G_1, r_j), G_{r_j,2}^c = \mathcal{G}^c(G_2, r_j), \cdots, G_{r_j,\ell}^c = \mathcal{G}^c(G_\ell, r_j)\}, \forall j = 1, 2, \ldots, k$. When the graph is attributed, we obtain the features for each node of the coarsened version of the graph by aggregating the features of the corresponding nodes in the original graph using simple aggregations (e.g., mean, max, sum), as commonly done in readout functions for GNNs [64].

**Regularizing the GNN.** Given the graph dataset $\mathcal{D}$, its coarsened versions $\mathcal{D}_{r_1}, \mathcal{D}_{r_2}, \cdots, \mathcal{D}_{r_k}$, and a GNN $f_\theta$, where $\theta$ are the parameters, we aim at regularizing the GNN so that the distribution of the node embeddings generated by the model for the graphs in the original dataset and their coarsened version is similar. In other words, for a given graph, we want the model to generate a *distribution* of node embeddings that is robust across different coarsened versions of the graph. In more detail, we regularize a GNN by minimizing the CMD [70] between the distribution of the node embeddings generated by the GNN for the original graph and the distribution of the node embeddings generated by the GNN for the coarsened version(s) of the graph. We choose CMD as it has proven to be successful and stable as a regularization term for non-linear models [42, 72]. Formally, let $\mathcal{L}$ be the supervised loss function that is used to train the model (e.g. cross entropy for classification), and let $\lambda$ be a term measuring the strength of the regularization. Our optimization problem is

$$\underset{\theta}{\arg\min} \, \mathcal{L} + \lambda \mathcal{L}_{\text{size}}, \text{ where } \mathcal{L}_{\text{size}} = \sum_{j=1}^{k} \sum_{i=1}^{\ell} \text{CMD}(f_\theta(G_i), f_\theta(G_{r_j,i}^c)). \tag{3}$$

Our method is model-agnostic, and can hence be applied to any GNN.

**Pseudocode and practical aspect.** Algorithm 1 presents the pseudocode for computing our regularization loss for a batch of graphs during training. The pseudocode is presented in an extended manner for clarity, but at a practical level, the coarsened versions of the training graphs are pre-computed before training, and the computation of the loss is done in a vectored manner so it is computed concurrently for all graphs in the batch in one pass (i.e., we do not iterate through the graphs in the batch). At a practical level, in our experiments we notice that training a model with our regularization introduces a $50\%$ overhead in the running time for a training epoch w.r.t. training the same model without regularization.

## 3.1 Limitations

Similarly to previous works [6, 68] we are assuming that there are some properties that determine the label of a graph and that do not depend on the size of the graph. In a scenario in which small graphs do not carry information that is relevant for solving the task on larger graphs, we do not expect our regularization to provide substantial benefits, and the best option would be to include larger graphs in the training set. In our experiments, the ratio between the average size of the graphs in the test set and the average size of the graphs in the training set is between 3 and 9. While our method shows significant performance improvements in this setting, it may show lower performance improvements when this ratio reaches much higher values.

---

**Algorithm 1** Computing Regularization Loss for an Input Batch during Training

---
**Require:** Coarsening ratios $C$, coarsening function $\mathcal{G}^c(\text{graph}, \text{ratio})$
**Input:** Batch $B$ of size $n_b$, GNN model $f_\theta$
  coarsened_batches $\leftarrow \{\}$
  **for** $r$ in $C$ **do**                               ▷ Create a new batch of coarsened graphs for each ratio
      Batch $B_r \leftarrow [\,]$
      **for** $G$ in $B$ **do**
         $G_r^c \leftarrow \mathcal{G}^c(G, r)$
         $B_r$.add$(G_r^c)$
      **end for**
      coarsened_batches $\leftarrow$ coarsened_batches $\cup$ $B_r$
  **end for**
  $\ell \leftarrow 0$                                           ▷ Initialize loss
  **for** $B_r$ in coarsened_batches **do**
      **for** $i$ in $\{1, 2, \ldots, n_b\}$ **do**
         embs_og $= f_\theta(B[i])$                  ▷ Compute node embeddings for original graph
         embs_coarse $= f_\theta(B_r[i])$     ▷ Compute node embeddings for coarsened version of graph
         $\ell \leftarrow \ell + \text{CMD}(\text{embs\_og}, \text{embs\_coarse})$     ▷ Compute CMD between node embeddings
      **end for**
  **end for**
  **Return** $\ell$

---

# 4 Analysis of Node Embeddings

Before evaluating how our regularization impacts the size-generalization performance of a model, we analyze the effects that our regularization has on the embeddings generated by the model. In order to do this, we consider two identical GIN [65] models and we train one with our regularization and one without. We then use these models to generate node embeddings and use the Central Kernel Alignment (CKA) [38] to study the generated representations, similarly to [34]. CKA takes as input two matrices $A \in \mathbb{R}^{m \times d'}, B \in \mathbb{R}^{m \times d''}$ of representations and provides a value between 0 and 1 quantifying how aligned the representations are (allowing for $d' \neq d''$). CKA quantifies the similarity of representations learned by (possibly) different models and gives us a way to study the effects of our regularization. We report results averaged over 10 different random seeds. We show only results for the DD dataset for space limitations, but the same trends are observed for the other datasets (see Appendix).

Table 1: Average CKA values between the node embeddings generated by two models, one trained with and one without our regularization, across the graphs in a dataset and their coarsened versions.

| Dataset | NCI1 | NCI109 | PROTEINS | DD |
|---|---|---|---|---|
| Original | $0.43 \pm 0.06$ | $0.58 \pm 0.10$ | $0.45 \pm 0.06$ | $0.47 \pm 0.01$ |
| Coarsened | $0.12 \pm 0.06$ | $0.38 \pm 0.13$ | $0.34 \pm 0.08$ | $0.40 \pm 0.01$ |

First, we ask if a model trained with our regularization and a model trained without our regularization produce similar node embeddings. To compare the node embeddings *between* the two models we obtain a representation for each graph by concatenating the node embeddings for that graph. We then compute the CKA between the representations generated by the model trained with regularization and the representations from model trained without regularization. We do this for the original graphs and the coarsened versions of the graphs, and report the average CKA. Results shown in Table 1 highlight that there is low alignment between the embeddings generated by a model trained with regularization and one trained without regularization (the average CKA across datasets is $0.48$ for the original graphs). This is even more apparent for the coarsened graphs, showing that our regularization

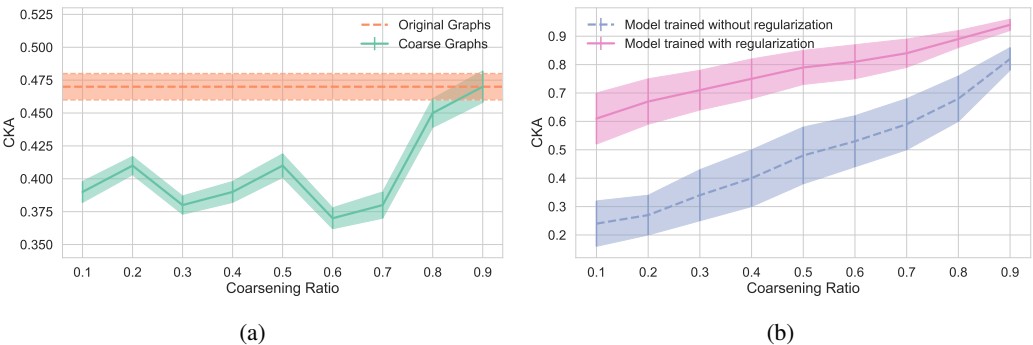

Figure 2: (a) Average CKA values between the node embeddings from two models, one trained with our regularization and one without, across the original graphs in the DD dataset and their coarsened versions for different ratios. The models produce representations which are not aligned (values are far from 1.0), specially for coarsened graphs. (b) Average CKA values between the node embeddings generated by the same model for the original graphs in DD and their coarsened versions. A model trained with our regularization produces representations of the coarsened graphs that are more aligned to the representations produced for the original graphs (i.e. it learns to be robust to size shifts).

is impacting the way a model generates embeddings, especially for size-shifted versions of the graphs. To analyze the trend more in detail, we also plot the CKA values across coarsening ratios in Figure 2 (a). Here we notice that the CKA between the embeddings generated by a model trained with our regularization and a model trained without, decreases significantly with ratios lower than 0.8. This shows that, while the alignment between the representations learned by the models trained with and without our regularization is low even for a coarsening ratio of 0.9 (average CKA value is 0.47), it becomes 15% lower (on average) for ratios smaller than 0.7, indicating that when the size shift is strong, there is a stronger misalignment between models trained with and without regularization.

Second, we ask if a model trained with our regularization is in fact being more robust to size shifts with respect to a model trained without our regularization. To answer this question we analyze the alignment between the representations generated by the same model, trained with or without regularization, on original graphs and coarsened versions of the graphs. In Figure 2 (b) we plot the average CKA between the node embeddings for the original graphs and the node embeddings for the coarsened versions of the graphs, both generated by the same model, for different ratios. The plot shows that a model trained with our regularization is learning to be more robust to size shifts, as its representation for coarsened versions of the graphs are much more aligned to the representations for the original graphs compared to a model trained without regularization. This is even more apparent for low coarsening ratios (i.e., when the shift is stronger), indicating that our regularization is in fact making the model more robust to size shifts.

## 5 Evaluation

**Datasets.** Following prior works [6, 68], we evaluate on datasets from the TUDataset library [48] with a train/test split explicitly designed to test for size-generalization. In particular, the training split contains the graphs with size smaller or equal than the 50-th percentile, and the test split contains graphs with a size larger or equal than the 90-th percentile. With this split, the average size of the test graphs is 3 to 9 times larger than the average size of the training graphs (in more detail, it is 3 for NCI1 and NCI109, 9 for PROTEINS, and 5 for DD). We use 10% of the training graphs as a validation set. More details on the datasets can be found in the Appendix.

**Models.** As in Bevilacqua et al. [6], we consider three standard GNN models: GCN [36], GIN [65], and PNA [15], a more expressive GNN: RPGNN [49], and two graph kernels: the Graphlet Counting kernel (GC Kernel) [58], and the WL Kernel [59]. We further apply the Invariant Risk Minimization (IRM) [1] penalty to the standard GNN models, and compare against the E-Invariant models ($\Gamma_{\text{1-hot}}, \Gamma_{\text{GIN}}, \Gamma_{\text{RPGIN}}$) introduced in Bevilacqua et al. [6]. For IRM, we follow [6], and

Table 2: Average Matthews Correlation Coefficient (MCC) for standard GNN models over the size-generalization test set. The models have been trained with (✓) and without (✗) our regularization method. The right-most column shows the average improvement brought by our regularization.

| Dataset | NCI1 | | NCI109 | | PROTEINS | | DD | | Avg. |
| Reg. | ✗ | ✓ | ✗ | ✓ | ✗ | ✓ | ✗ | ✓ | Impr. |
|---|---|---|---|---|---|---|---|---|---|
| PNA | $0.19 \pm 0.08$ | $0.22 \pm 0.07$ | $0.23 \pm 0.07$ | $0.24 \pm 0.07$ | $0.22 \pm 0.12$ | $0.33 \pm 0.09$ | $0.23 \pm 0.09$ | $0.27 \pm 0.08$ | **+22%** |
| GCN | $0.17 \pm 0.06$ | $0.25 \pm 0.06$ | $0.15 \pm 0.06$ | $0.19 \pm 0.06$ | $0.21 \pm 0.10$ | $0.29 \pm 0.13$ | $0.24 \pm 0.07$ | $0.26 \pm 0.07$ | **+30%** |
| GIN | $0.19 \pm 0.06$ | $0.23 \pm 0.08$ | $0.18 \pm 0.05$ | $0.20 \pm 0.05$ | $0.25 \pm 0.07$ | $0.36 \pm 0.11$ | $0.23 \pm 0.09$ | $0.25 \pm 0.09$ | **+21%** |

Table 3: Comparison of average Matthews Correlation Coefficient (MCC) over the size-generalization test data between the standard GNN models trained with our proposed regularization and previously proposed methods. Highlighted are the **first**, **second**, and **third** best performing models per dataset.

| Dataset | NCI1 | NCI109 | PROTEINS | DD |
|---|---|---|---|---|
| PNA (reg) [ours] | $0.22 \pm 0.07$ | $0.24 \pm 0.07$ | $0.33 \pm 0.09$ | $0.27 \pm 0.08$ |
| GCN (reg) [ours] | $0.25 \pm 0.06$ | $0.19 \pm 0.06$ | $0.29 \pm 0.13$ | $0.26 \pm 0.07$ |
| GIN (reg) [ours] | $0.23 \pm 0.08$ | $0.20 \pm 0.05$ | $0.36 \pm 0.11$ | $0.25 \pm 0.09$ |
| PNA + IRM [1] | $0.17 \pm 0.07$ | $0.20 \pm 0.07$ | $0.21 \pm 0.12$ | $0.24 \pm 0.08$ |
| GCN + IRM [1] | $0.22 \pm 0.06$ | $0.20 \pm 0.06$ | $0.23 \pm 0.16$ | $0.23 \pm 0.08$ |
| GIN + IRM [1] | $0.18 \pm 0.06$ | $0.15 \pm 0.04$ | $0.24 \pm 0.08$ | $0.21 \pm 0.10$ |
| WL kernel [59] | $0.39 \pm 0.00$ | $0.21 \pm 0.00$ | $0.00 \pm 0.00$ | $0.00 \pm 0.00$ |
| GC kernel [58] | $0.02 \pm 0.00$ | $0.01 \pm 0.00$ | $0.29 \pm 0.00$ | $0.00 \pm 0.00$ |
| RPGIN [49] | $0.18 \pm 0.06$ | $0.16 \pm 0.04$ | $0.22 \pm 0.08$ | $0.13 \pm 0.04$ |
| $\Gamma_{1\text{-hot}}$ [6] | $0.15 \pm 0.05$ | $0.22 \pm 0.06$ | $0.18 \pm 0.08$ | $0.22 \pm 0.09$ |
| $\Gamma_{GIN}$ [6] | $0.24 \pm 0.05$ | $0.16 \pm 0.07$ | $0.28 \pm 0.10$ | $0.27 \pm 0.05$ |
| $\Gamma_{RPGIN}$ [6] | $0.26 \pm 0.05$ | $0.19 \pm 0.06$ | $0.26 \pm 0.07$ | $0.20 \pm 0.05$ |

artificially create two environments: one with graphs with size smaller than the median size of the graphs in the training set, and one with graphs with size larger than the median size in the training set.

**Hyperparameters and Evaluation Protocol.** For the standard GNN models we adopt the same hyperparameters of Bevilacqua et al. [6], which were obtained from a tuning procedure involving number of layers, learning rate, batch size, hidden layers dimension, and regularization terms. All models are trained with early stopping (i.e., taking the weights at the epoch with best performance on the validation set). We do not modify the original hyperparameters when we apply our regularization. We tune the regularization coefficient $\lambda$ and the coarse ratios $C$ for GIN on the PROTEINS dataset (we found $\lambda = 0.1$ and $C = \{0.8, 0.9\}$ to be the best on the validation set), and we apply these settings to *all models and datasets* when using our regularization, to show that our method can work without extensive (and expensive) hyperparameter tuning. We use the SGC coarsening algorithm [32] to obtain the coarsened versions of the graphs (chosen for its theoretical properties). The only hyperparameter that is tuned on a per-dataset basis is the aggregation strategy used to obtain the features for the nodes in the coarsened versions of the graphs (in particular we take the best performing between 'sum', 'max', and 'mean'). For all other models we use the hyperparameters introduced by their respective papers. Bevilacqua et al. [6] presented results by averaging over 10 runs (each time with a different random seed), but given the high variance observed, we re-run the experiments for all the models and present the results averaged over 50 runs. More details on the hyperparameters can be found in the Appendix and source code is publicly available[1].

## 5.1 Results

Table 2 shows how the three considered standard GNN models perform on the test set, containing the 10% largest graphs in the dataset, when trained with and without our regularization on the 50% smallest graphs in the dataset. As this split leads to an imbalanced dataset, we follow previous works and report the results in terms of Matthews correlation coefficient (MCC), which has been shown

---

[1]`https://github.com/DavideBuffelli/SizeShiftReg`

to be more reliable in imbalanced settings with respect to other common metrics [14]. MCC gives a value between $-1$ and 1, where $-1$ indicates perfect disagreement, 0 is the value for a random guesser, and 1 indicates perfect agreement between the predictions and the true labels. The results show that the use of the proposed regularization is always beneficial to the performance of the models, and that it leads to an average improvement across datasets of **21 to 30%**.

In Table 3, we compare the standard GNNs trained with our regularization strategy against more expressive models (RPGNN), graph kernels, models trained with the IRM strategy, and the E-invariant models. We notice that on 3 out of 4 datasets, a "standard" GNN trained with our regularization obtains the best performance on the test set composed of graphs with size larger than those present in the training set. Furthermore on all datasets there is at least one model trained with our regularization in the top-3 best performing models. We also confirm previous results [6, 17] showing that IRM is not effective in the graph domain, and that using theoretically more expressive

Table 4: Average MCC results on the Deezer dataset for models trained with (✓) and without (✗) our regularization.

| Dataset | Deezer | |
|---|---|---|
| Reg. | ✗ | ✓ |
| PNA | $0.59 \pm _{0.06}$ | $0.64 \pm _{0.07}$ |
| GCN | $0.49 \pm _{0.10}$ | $0.59 \pm _{0.06}$ |
| GIN | $0.55 \pm _{0.08}$ | $0.61 \pm _{0.07}$ |

models, like RPGNN, does not necessarily lead to good size-generalization performance. Graph kernels are highly dataset-dependent, and, while they can perform well for some datasets, in many cases they fail to perform better than a random classifier. Perhaps more surprisingly, on 3 out of 4 datasets there is at least one "standard" GNN trained with our regularization that performs comparably or better than the best E-Invariant model. In fact, E-invariant models are tied to the assumed causal model for the generation of graphs, which is not guaranteed to hold reliably for real-world datasets for which we do not have this kind of information. Futhermore, E-invariant models require the computation of induced homomorphism densities over all possible connected k-vertex subgraphs both at training time and at test time. Our method instead tries to *learn* to be robust to size-shifts, does not require any additional computation at inference time, and yet can lead even simple models like GCN to have size-generalization performance comparable to E-invariant models.

Finally, in addition to the benchmarks considered by Bevilacqua et al. [6], we also experiment on a dataset from a different domain. In particular, we consider a social network dataset obtained from Deezer [55]. We follow the same evaluation strategy as before: train on the $50\%$ smallest and test on the $10\%$ largest, and we apply our regularization with $\lambda = 0.1$ and $C = \{0.8, 0.9\}$ without any additional hyperparameter tuning. Results shown in Table 4 confirm the effectiveness of our method.

## 5.2 Ablation Study

In this section we use a PNA model to study the contribution and importance of the different components of our regularization. We choose PNA because of its performance in the previous analysis, but similar conclusions are observed for GIN and GCN (as we report in the Appendix).

**Changing Size of Coarsened Graphs.** In Table 5 we train a PNA model with our regularization strategy using different coarsening ratios $C$. We notice an overall trend in which the performance decreases as the coarsening ratio decreases. This follows intuitively as very low coarsening ratios may lead to uninformative graphs. Furthermore we notice that using all ratios (from 0.1 to 0.9) is usually not effective and setting $C = \{0.8, 0.9\}$ seems a good default option, leading to the best performance on 3 out of 4 datasets.

**Changing Coarsening Method.** In Figure 3 we show how the performance of a PNA model change when it is regularized with our method using graphs coming from different coarsening functions. We consider Spectral Graph Coarsen-

Table 5: Average MCC results on the size-generalization test set for a PNA model trained with our regularization strategy using different coarsening ratios.

| Datasets Ratio(s) | NCI1 | NCI109 | PROTEINS | DD |
|---|---|---|---|---|
| 0.1 | $0.07 \pm _{0.11}$ | $0.19 \pm _{0.08}$ | $0.12 \pm _{0.15}$ | $0.07 \pm _{0.14}$ |
| 0.2 | $0.11 \pm _{0.12}$ | $0.20 \pm _{0.08}$ | $0.18 \pm _{0.16}$ | $0.20 \pm _{0.11}$ |
| 0.3 | $0.15 \pm _{0.09}$ | $0.22 \pm _{0.07}$ | $0.17 \pm _{0.16}$ | $0.22 \pm _{0.10}$ |
| 0.4 | $0.18 \pm _{0.08}$ | $0.22 \pm _{0.08}$ | $0.23 \pm _{0.14}$ | $0.26 \pm _{0.11}$ |
| 0.5 | $0.22 \pm _{0.08}$ | $0.24 \pm _{0.07}$ | $0.29 \pm _{0.12}$ | $0.22 \pm _{0.09}$ |
| 0.6 | $0.23 \pm _{0.08}$ | $0.23 \pm _{0.07}$ | $0.26 \pm _{0.09}$ | $0.27 \pm _{0.10}$ |
| 0.7 | $0.22 \pm _{0.07}$ | $0.24 \pm _{0.06}$ | $0.30 \pm _{0.14}$ | $0.24 \pm _{0.06}$ |
| 0.8 | $0.24 \pm _{0.06}$ | $0.23 \pm _{0.07}$ | $0.32 \pm _{0.11}$ | $0.25 \pm _{0.09}$ |
| 0.9 | $0.19 \pm _{0.06}$ | $0.21 \pm _{0.08}$ | $0.28 \pm _{0.12}$ | $0.25 \pm _{0.10}$ |
| $\{0.1, 0.9\}$ | $0.12 \pm _{0.10}$ | $0.20 \pm _{0.06}$ | $0.14 \pm _{0.16}$ | $0.14 \pm _{0.14}$ |
| $\{0.5, 0.9\}$ | $0.23 \pm _{0.08}$ | $0.22 \pm _{0.07}$ | $0.32 \pm _{0.12}$ | $0.27 \pm _{0.08}$ |
| $\{0.8, 0.9\}$ | $0.22 \pm _{0.07}$ | $0.24 \pm _{0.07}$ | $0.33 \pm _{0.09}$ | $0.27 \pm _{0.08}$ |
| $\{0.3, 0.7\}$ | $0.22 \pm _{0.09}$ | $0.21 \pm _{0.08}$ | $0.24 \pm _{0.11}$ | $0.25 \pm _{0.09}$ |
| ALL | $0.18 \pm _{0.09}$ | $0.18 \pm _{0.09}$ | $0.17 \pm _{0.14}$ | $0.23 \pm _{0.10}$ |

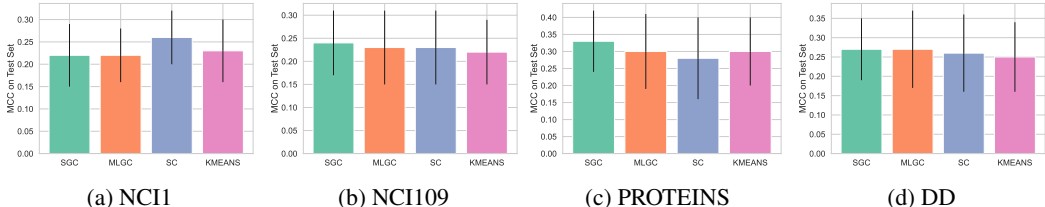

| (a) NCI1 | (b) NCI109 | (c) PROTEINS | (d) DD |

Figure 3: Average MCC on the test set of a PNA model trained with our regularization strategy using different coarsening methods: Spectral Graph Coarsening (SGC), Multi-level Graph Coarsening (MLGC), Spectral Clustering (SC), K-Means (KMEANS).

ing (SGC) and Multi-level Graph Coarsening (MLGC) introduced by Jin et al. [32], and two baselines, referred to as SC and KMEANS, based on standard spectral clustering [28] and K-Means clustering [40]. For the baselines we apply the clustering algorithm, merge the nodes belonging to the same cluster into a single super-node, and create an edge between two super-nodes if there is at least one edge in the original graph going from a node in one cluster to the other. For all coarsening methods we use the same coarsening ratios as above (i.e., $C = \{0.8, 0.9\}$), and we apply the same training procedure and regularization weight $\lambda$, keeping all hyperparameters equal across methods. We notice that our regularization strategy is quite robust towards the choice of the coarsening method, but, on average, specialised methods like SGC and MLGC tend to provide the best results.

## 6 Related Work

**Graph neural networks and size-generalization.** Since the first instances of graph neural networks have appeared [60, 57], the field has exploded in recent years with a plethora of different architectures (e.g., [36, 65, 15, 62]), applications (e.g., [69, 67, 37, 23, 8]), and studies on their expressivity and properties (e.g., [65, 44, 47, 9]). For a complete review of the field, we refer to the excellent surveys by Chami et al. [12] and by Wu et al. [64]. Many works have noticed poor size-generalization in GNNs [33, 25, 6, 68]. For instance, Joshi et al. [33] consider the task of learning solutions for the traveling salesman problem, and show that GNNs struggle at generalizing to graphs larger than those appearing in the training set. Similarly, Gasteiger et al. [25] observe that GNNs applied to the molecular domain can be very sensible to several shifts in the data, including shifts in the size of the graphs. Some works have reported good size-generalization by creating ad-hoc models that leverage the properties of the domain [2, 3, 56, 61, 22]. For example, Tsubaki and Mizoguchi [61] impose constraints dictated by the physical properties of their data, and report good size-generalization in predicting the energy of a molecule. While these works show that placing the right inductive biases can help the size-generalization capabilities of GNNs, the current literature is lacking a general method that can be applied on generic GNNs.

Bevilacqua et al. [6] tackle the issue of poor size-generalization by assuming a causal model describing the generative process for the graphs in the dataset, and designing a specific model which can be invariant to the size of the graphs obtained through this causal model. While this strategy shows great size-generalization capabilities on synthetic graphs generated according to their causal model, its benefits decrease when applied to real-world graphs where there are no guarantees that the causal model is correct. This method requires an ad-hoc model and additional computations at both train and test time, while our method can be applied on *any* GNN and does not require pre-computations at test time. Yehudai et al. [68] provide a theoretical and empirical analysis showing that, even for simple tasks, it is not trivial to obtain a GNN with good size-generalization properties. Yehudai et al. [68] consider a different scenario from ours, as, together with the labelled training graphs, they assume access to graphs from the test distribution, and treat the task as a domain-adaptation scenario.

**Graph coarsening.** While there is no consensus on what is the best metric to consider for coarsening a graph, many methods and metrics have been proposed in recent years [45, 43, 18, 7, 32]. There have also been some attempts at using GNNs for the task of graph coarsening [10, 46]. Orthogonally, graph coarsening has been used to reduce the computational resources needed for training GNNs on large graphs [71, 31]. For a detailed presentation of graph coarsening we refer to Chen et al. [13].

**Invariant risk minimization, domain adaptation, and regularization with discrepancy measures.** Our method is conceptually related to Invariant Risk Minimization (IRM) [1], which aims at learning representations that are invariant across training environments (where different training environments are intended as sets of data points collected under different conditions). IRM has been shown to have several shortcomings, specially when the data comes from a single environment (e.g. when there is access only to small graphs) [54, 6]. Our method does not require access to different training environments, which may not be easy to obtain in practical scenarios, and our results show that, as also observed in [6, 17], IRM seems ineffective for improving size-generalization in GNNs.

Similarly to IRM, the field of domain adaptation [4] subsumes access to (unlabelled) external data, in addition to labelled training data, and aims at transferring knowledge between domains. Domain adaptation has been used in a large variety of fields, and many surveys are available [16, 63, 27]. Ding et al. [17] analyze how existing domain adaptation algorithms perform on graph data, and observe that these tend to not be effective, indicating that new methods specific for graph data are needed.

Discrepancy measures that have been used for regularizing deep learning models are the Maximum mean discrepancy (MMD) [19, 41, 66, 42] and the central moment discrepancy (CMD) [52, 70, 72], which has been shown to be empirically more effective. We further mention the work by Zhu et al. [72] which, similarly to ours, uses CMD to obtain a GNN that is robust to biases in the sampling process that is used to select the nodes for the training set in node classification scenarios. Finally, we mention the work by Joshi et al. [34] which uses CMD to study the representations of GNNs.

## 7    Conclusions

GNNs are heavily used models for the task of graph classification. In many domains it is typical for graphs to vary in size, and while GNNs are designed to be able to process graphs of any size, empirical results across literature have highlighted that GNNs struggle at generalizing to sizes unseen during training. In this work we introduce a regularization strategy that can be applied on any GNN, and that improves size-generalization performance from smaller to larger graphs by up to 30%.

## Acknowledgments and Disclosure of Funding

The authors thank Chaitanya Joshi and Iulia Duta for the great feedback provided on earlier versions of the paper.

This work is supported, in part, by the Italian Ministry of Education, University and Research (MIUR), under PRIN Project n. 20174LF3T8 "AHeAD" and the initiative "Departments of Excellence" (Law 232/2016), and by University of Padova under project "SID 2020: RATED-X".

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
