# Appendix for the Paper: "SizeShiftReg: a Regularization Method for Improving Size-Generalization in Graph Neural Networks"

**Davide Buffelli**
Department of Information Engineering
University of Padova
Padova, Italy 35131
davide.buffelli@unipd.it

**Pietro Liò**
Department of Computer Science and Technology
University of Cambridge
Cambridge, United Kingdom CB3 0FD
pietro.lio@cl.cam.ac.uk

**Fabio Vandin**
Department of Information Engineering
University of Padova
Padova, Italy 35131
fabio.vandin@unipd.it

## Appendix

This Appendix contains the following additional material. In Section A we report more statistics regarding the considered datasets. In Section B we report the results for the CKA analysis on the other datasets not shown in the main paper for space limitations. In Section C we report more details on the hyperparameters of the models used for our experiments. In Section D we report the results for the ablation study on the other models not shown in the main paper for space limitations. In Section E we show the overhead (in terms of training time) incurred by our method. In Section F we report information on the computing resources used to perform our experimental results, links to access the datasets, and the licence of the publicly available libraries used in our code.

## A    Dataset Information

Table 1: Dataset statistics, this table is taken from Yehudai et al. [13], Bevilacqua et al. [1].

| | NCI1 | | | NCI109 | | |
|---|---|---|---|---|---|---|
| | ALL | SMALLEST 50% | LARGEST 10% | ALL | SMALLEST 50% | LARGEST 10% |
| CLASS A | 49.95% | 62.30% | 19.17% | 49.62% | 62.04% | 21.37% |
| CLASS B | 50.04% | 37.69% | 80.82% | 50.37% | 37.95% | 78.62% |
| NUM OF GRAPHS | 4110 | 2157 | 412 | 4127 | 2079 | 421 |
| AVG GRAPH SIZE | 29 | 20 | 61 | 29 | 20 | 61 |

| | PROTEINS | | | DD | | |
|---|---|---|---|---|---|---|
| | ALL | SMALLEST 50% | LARGEST 10% | ALL | SMALLEST 50% | LARGEST 10% |
| CLASS A | 59.56% | 41.97% | 90.17% | 58.65% | 35.47% | 79.66% |
| CLASS B | 40.43% | 58.02% | 9.82% | 41.34% | 64.52% | 20.33% |
| NUM OF GRAPHS | 1113 | 567 | 112 | 1178 | 592 | 118 |
| AVG GRAPH SIZE | 39 | 15 | 138 | 284 | 144 | 746 |

In Table 1 (taken from [13, 1]) we report the information about the graphs in the considered datasets.

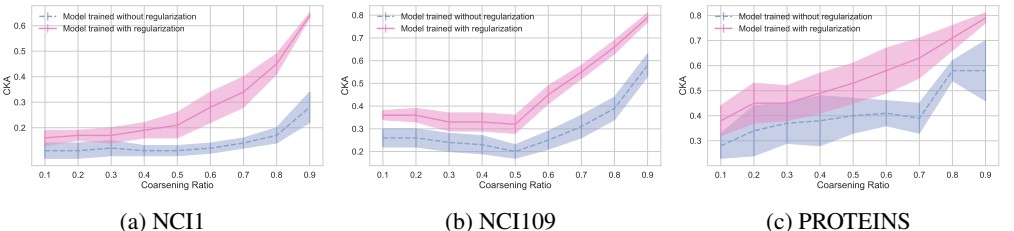

|            |            |              |
|:----------:|:----------:|:------------:|
| (a) NCI1   | (b) NCI109 | (c) PROTEINS |

Figure 1: Average CKA values between the node embeddings generated by the same model for the original graphs in (a) NCI1, (b) NCI109, (c) PROTEINS, and their coarsened versions. A model trained with our regularization produces representations of the coarsened graphs that are more aligned to the representations produced for the original graphs (i.e. it learns to be robust to size shifts).

## B    Embedding Analysis Results for Other Datasets

In Figure 1 we show the continuation of Figure 2 (b) from the main paper, including plots on all datasets. In more detail, we report plots for the average CKA [7] values between the node embeddings for the original graphs and the node embeddings for the coarsened versions of the graphs, both generated by the same model, for different ratios. The plot shows that a model trained with our regularization is learning to be more robust to size shifts, as its representation for coarsened versions of the graphs are much more aligned (higher CKA values) to the representations for the original graphs compared to a model trained without regularization. This is even more apparent for low coarsening ratios (i.e., when the shift is stronger), indicating that our regularization is in fact making the model more robust to size shifts. The reason why the CKA values for a model trained with and without regularization tend to become similar for small ratios (which is instead not apparent on the DD dataset shown in Figure 2 (b) in the main paper) is that the graphs in NCI1, NCI109, and PROTEINS are much smaller than the graphs in DD (see Section A), and hence graphs tend to become completely uninformative with ratios lower than $0.5$.

## C    Hyperparameters and Evaluation Procedure

**Hyperparameters.**    As we follow exactly the same procedures as previous work [13, 1], we refer the reader to those papers for an in depth discussion, and we report below the description of the standard GNN models used in our experiments (which follows from the above mentioned papers).

The GCN[6], GIN [12], and PNA [2] models used in our experiments have 3 graph convolution layers and a final multi-layer perceptron with a softmax activation function to obtain the predictions. Batch norm is used in between graph convolution layers, and ReLU is used as activation function. The networks are trained with a dropout of $0.3$, and were tuned using the validation set. In particular the batch size was chosen between $64$ and $128$, the learning rate between $0.01, 0.005$, and $0.001$, and the network width between $32$ and $64$. All models are trained with early stopping (taking the wights related to the epoch with the highest MCC on the validation set), and the different classes were weighted in the supervised loss function according to the frequency of a class in the training data (to deal with the imbalance in the data, as explained in the main paper).

When applying our method in Section 3 and in Table 2 and Table 3 in the main paper, we use $\lambda = 0.1$ and $C = (0.8, 0.9)$ for all models and datasets. These parameters were taken from a tuning procedure on the validation set described below.

**Tuning and Evaluation Procedure.** To identify the values of $\lambda$ and $C$ to use for our method, we tried different values ($\lambda = \{1.0, 0.1, 0.01, 0.001\}$ and $C = \{(0.5), (0.8), (0.9), (0.5, 0.8), (0.5, 0.9), (0.8, 0.9)\}$) on the validation set using a GIN model on the PROTEINS dataset. We found that $\lambda = 0.1$ and $C = (0.8, 0.9)$ performed best on the validation set. As the results on the validation sets were good (i.e., they were leading to better performance with respect to a model trained without regularization), we adopted the same values of $\lambda$ and $C$ for all datasets and models to show that our method can work without extensive (and expensive)

Table 2: Table shows mean (standard deviation) Matthews correlation coefficient (MCC) for a GCN model trained with our proposed regularization with the Spectral Graph Coarsening (SGC) strategy applied with different coarsening ratios.

| Datasets Ratio | NCI1 | NCI109 | PROTEINS | DD |
|---|---|---|---|---|
| 0.1 | $0.09 \pm 0.12$ | $0.23 \pm 0.07$ | $0.05 \pm 0.14$ | $0.15 \pm 0.09$ |
| 0.2 | $0.17 \pm 0.11$ | $0.23 \pm 0.07$ | $0.05 \pm 0.16$ | $0.20 \pm 0.08$ |
| 0.3 | $0.18 \pm 0.10$ | $0.23 \pm 0.07$ | $0.08 \pm 0.18$ | $0.23 \pm 0.07$ |
| 0.4 | $0.21 \pm 0.07$ | $0.24 \pm 0.06$ | $0.09 \pm 0.17$ | $0.24 \pm 0.09$ |
| 0.5 | $0.23 \pm 0.07$ | $0.22 \pm 0.05$ | $0.21 \pm 0.14$ | $0.27 \pm 0.09$ |
| 0.6 | $0.24 \pm 0.05$ | $0.21 \pm 0.06$ | $0.23 \pm 0.17$ | $0.25 \pm 0.08$ |
| 0.7 | $0.26 \pm 0.06$ | $0.21 \pm 0.05$ | $0.25 \pm 0.16$ | $0.24 \pm 0.08$ |
| 0.8 | $0.27 \pm 0.06$ | $0.20 \pm 0.04$ | $0.29 \pm 0.15$ | $0.27 \pm 0.08$ |
| 0.9 | $0.25 \pm 0.07$ | $0.19 \pm 0.05$ | $0.22 \pm 0.14$ | $0.25 \pm 0.09$ |
| 0.1-0.9 | $0.15 \pm 0.11$ | $0.21 \pm 0.06$ | $0.09 \pm 0.15$ | $0.21 \pm 0.09$ |
| 0.5-0.9 | $0.22 \pm 0.05$ | $0.22 \pm 0.06$ | $0.28 \pm 0.12$ | $0.25 \pm 0.08$ |
| 0.8-0.9 | $0.25 \pm 0.06$ | $0.19 \pm 0.06$ | $0.29 \pm 0.13$ | $0.26 \pm 0.07$ |
| 0.3-0.7 | $0.21 \pm 0.07$ | $0.22 \pm 0.05$ | $0.10 \pm 0.14$ | $0.23 \pm 0.07$ |
| ALL | $0.17 \pm 0.08$ | $0.23 \pm 0.09$ | $0.12 \pm 0.16$ | $0.21 \pm 0.09$ |

Table 3: Table shows mean (standard deviation) Matthews correlation coefficient (MCC) for a GIN model trained with our proposed regularization with the Spectral Graph Coarsening (SGC) strategy applied with different coarsening ratios.

| Datasets Ratio | NCI1 | NCI109 | PROTEINS | DD |
|---|---|---|---|---|
| 0.1 | $0.07 \pm 0.11$ | $0.04 \pm 0.11$ | $0.26 \pm 0.12$ | $0.22 \pm 0.10$ |
| 0.2 | $0.07 \pm 0.11$ | $0.07 \pm 0.11$ | $0.30 \pm 0.10$ | $0.24 \pm 0.12$ |
| 0.3 | $0.07 \pm 0.11$ | $0.10 \pm 0.09$ | $0.29 \pm 0.15$ | $0.25 \pm 0.09$ |
| 0.4 | $0.02 \pm 0.13$ | $0.09 \pm 0.09$ | $0.31 \pm 0.15$ | $0.24 \pm 0.09$ |
| 0.5 | $0.07 \pm 0.11$ | $0.10 \pm 0.11$ | $0.32 \pm 0.12$ | $0.25 \pm 0.09$ |
| 0.6 | $0.02 \pm 0.11$ | $0.08 \pm 0.07$ | $0.34 \pm 0.12$ | $0.26 \pm 0.10$ |
| 0.7 | $0.01 \pm 0.09$ | $0.09 \pm 0.06$ | $0.31 \pm 0.14$ | $0.25 \pm 0.11$ |
| 0.8 | $0.15 \pm 0.10$ | $0.15 \pm 0.07$ | $0.36 \pm 0.11$ | $0.27 \pm 0.08$ |
| 0.9 | $0.22 \pm 0.08$ | $0.19 \pm 0.05$ | $0.34 \pm 0.10$ | $0.25 \pm 0.10$ |
| 0.1-0.9 | $0.09 \pm 0.10$ | $0.05 \pm 0.09$ | $0.32 \pm 0.15$ | $0.22 \pm 0.10$ |
| 0.5-0.9 | $0.03 \pm 0.10$ | $0.09 \pm 0.07$ | $0.32 \pm 0.13$ | $0.23 \pm 0.09$ |
| 0.8-0.9 | $0.23 \pm 0.08$ | $0.20 \pm 0.05$ | $0.36 \pm 0.11$ | $0.25 \pm 0.09$ |
| 0.3-0.7 | $0.05 \pm 0.12$ | $0.08 \pm 0.09$ | $0.26 \pm 0.15$ | $0.24 \pm 0.10$ |
| ALL | $0.05 \pm 0.12$ | $0.11 \pm 0.09$ | $0.25 \pm 0.16$ | $0.23 \pm 0.10$ |

Table 4: Table shows mean (standard deviation) Matthews correlation coefficient (MCC) for a GCN model trained with our proposed regularization and different coarsening strategies: Spectral Clustering (SC), Spectral Graph Coarsening (SGC), Multilevel Graph Coarsening (MLGC), K-Means (KMEANS).

| Datasets | NCI1 | NCI109 | PROTEINS | DD |
|---|---|---|---|---|
| GCN-SGC | $0.25 \pm 0.06$ | $0.19 \pm 0.06$ | $0.29 \pm 0.13$ | $0.26 \pm 0.07$ |
| GCN-MLGC | $0.21 \pm 0.07$ | $0.21 \pm 0.06$ | $0.27 \pm 0.14$ | $0.25 \pm 0.06$ |
| GCN-SC | $0.28 \pm 0.07$ | $0.18 \pm 0.06$ | $0.24 \pm 0.16$ | $0.24 \pm 0.07$ |
| GCN-KMEANS | $0.28 \pm 0.06$ | $0.18 \pm 0.05$ | $0.25 \pm 0.15$ | $0.24 \pm 0.07$ |

Table 5: Table shows mean (standard deviation) Matthews correlation coefficient (MCC) for a GIN model trained with our proposed regularization and different coarsening strategies: Spectral Clustering (SC), Spectral Graph Coarsening (SGC), Multilevel Graph Coarsening (MLGC), K-Means (KMEANS).

| Datasets | NCI1 | NCI109 | PROTEINS | DD |
|---|---|---|---|---|
| GIN-SGC | $0.23 \pm 0.08$ | $0.20 \pm 0.05$ | $0.36 \pm 0.11$ | $0.25 \pm 0.09$ |
| GIN-MLGC | $0.22 \pm 0.07$ | $0.19 \pm 0.06$ | $0.36 \pm 0.10$ | $0.25 \pm 0.10$ |
| GIN-SC | $0.08 \pm 0.11$ | $0.07 \pm 0.08$ | $0.32 \pm 0.12$ | $0.21 \pm 0.10$ |
| GIN-KMEANS | $0.21 \pm 0.09$ | $0.20 \pm 0.06$ | $0.35 \pm 0.08$ | $0.28 \pm 0.09$ |

hyperparameter tuning. It is possible that dataset-specific and model-specific tuning can lead to higher results.

We first obtained the results for Table 1, Figure 2, Table 2, and Table 3, in the main paper. Then, only *afterwards*, we performed the ablation study. The ablation is used to understand the impact of the components of our method only after having evaluated it, as is the standard procedure for ablation studies.

# D   Full Ablation Study Results

We report the results of the ablation study also for the GIN and GCN models.

**Changing Size of Coarsened Graphs.**   We report in Table 2 and Table 3 the performance of a GCN [6] and GIN [12] model when trained with our regularization strategy with different coarsening ratios $C_r$. All other hyperparameters are kept the same as for the main results in our paper. As for the PNA model shown in the main paper, we notice an overall trend in which the performance decreases as the coarsening ratio decreases. This follows intuitively as very low coarsening ratios may lead to uninformative graphs. Furthermore we notice that using all ratios (from 0.1 to 0.9) is usually not effective and setting $C = \{0.8, 0.9\}$ seems a good default option.

**Changing Coarsening Method.**   In Table 4 and Table 5 we report the performance of a GCN [6] and a GIN [12] model trained with our regularization using different coarsening functions. All other parameters are kept the same. As for the PNA shown in the main paper we notice that our method is quite robust to the choice of the coarsening function. However, in most cases, specialised methods like SGC and MLGC provide the best results.

Table 6: Average (over ten runs) time (s) for performing one epoch during training of standard GNN models trained with (✓) and without (✗) our regularization method. Standard deviation is not reported as it is small and similar for all results.

| Dataset | NCI1 | | NCI109 | | PROTEINS | | DD | |
| Reg. | ✗ | ✓ | ✗ | ✓ | ✗ | ✓ | ✗ | ✓ |
| --- | --- | --- | --- | --- | --- | --- | --- | --- |
| PNA | 2.02 | 3.1 | 3.21 | 4.94 | 2.33 | 3.56 | 3.45 | 4.88 |
| GCN | 1.92 | 2.91 | 2.14 | 3.88 | 2.29 | 3.41 | 3.26 | 4.74 |
| GIN | 1.95 | 3.02 | 2.15 | 3.86 | 2.3 | 3.42 | 3.41 | 4.75 |

## E   Training Times

Table 6 shows the time in seconds for training a model with and without our regularization. We notice that our method introduces an average 50% overhead in training time.

## F   Compute and Licence Information

All our experiments were performed on a machine with a Nvidia 1080Ti GPU and a CPU cluster equipped with 8 CPUs 12-core Intel Xeon Gold 5118@2.30GHz with 1.5Tb of RAM.

The TUDataset [9] we used for our experiments are publicly available online[1]. Our code makes use of the following publicly available libraries: PyTorch [10] (BSD-Style License), PyTorch Geometric [4] (MIT License), PyTorch Lightning [3] (Apache-2.0 license), scikit-learn [11] (BSD-3 license), numpy [5] (BSD-3 license), ray[tune] [8] (Apache 2.0).