# OpenReview forum: "SizeShiftReg: a Regularization Method for Improving Size-Generalization in Graph Neural Networks"
_NeurIPS.cc/2022/Conference — NeurIPS 2022 Accept_

### Official Review · Reviewer_k3VB · 2022-06-23

**Rating:** 7
**Confidence:** 4
**Soundness:** 3 good
**Presentation:** 4 excellent
**Contribution:** 3 good

**Summary:**

This paper presents a simple and intuitive regularization technique to make graph networks generalize better to graphs with different sizes, by employing coarsened representations of the same graph and encouraging alignment of node embeddings between such graphs.

**EDIT after rebuttal:**

I thank the authors for clarifying important aspects of the evaluation procedure. It appears my assumptions were incorrect. I therefore recommend that the paper is accepted, as it provides a simple, clear, novel and effective mechanism to tackle an interesting problem for the graph machine learning community, something which is rare these days.

**Questions:**

- Section 4: we have no details about the experimental procedure used here. Which lambda values have been used here to produce Table 1? Maybe it would be better to include this analysis in Section 5, so as to improve the overall organization of the paper.
- How does the distribution over the sizes of the graphs changes after pre-processing? This analysis could be interesting in datasets where graphs' sizes greatly vary, e.g. social datasets from TUDataset benchmarks page.
- From Section 3.1., why not trying a single experiment with much higher coarsening ratios? This could be a nice addition to the paper. Social datasets could be a good option here as well.

**Limitations:**

Limitations of the paper mainly lie in the experimental results, which may be invalid if my argument is correct.

**Strengths And Weaknesses:**

It is rare to find and review a paper which clearly presents a simple, intuitive and sensible idea to this kind of conferences. Overall, it was a pleasure to read and very clear under all aspects. The paper seems technically sound, although it is not clear how the specific choice of CKA influences the results, as it compares graph representations of original and coarsened graphs which have different dimensions (if my understanding is correct). The underlying motivations of the paper is clear and the proposed solution seems simple enough to be used in everyday research/industrial contexts. Perhaps it is not very clear what is the computational complexity of the regularization loss, and the authors may want to consider Section 4 after the experimental setup is introduced (see my first question below).

Despite the paper being of high quality in many respects, I have serious concerns about the experimental protocol which led to the creation of Table 3, the main empirical evaluation of the paper. From lines 219-234, it emerges that most hyper-parameters where chosen from [6], which is reasonable as long as the empirical setup stays the very same.

What appears definitely less reasonable and fundamentally wrong seems to be:
- using the same settings of $\lambda$ and $C$ for all datasets and models after looking at a single dataset (see lines 224-225) and Table 3 in the appendix B
- lack of information, either in the appendix or in the main paper, about the values of $\lambda$ cross-validated against the $\textbf{validation}$ set
- Evidence, by looking at Table 4 (caption, main paper) and Tables 2,3 (appendix B) that the hyper-parameters have been set after looking at the MCC performances on the $\textbf{test}$ set. This would mean that the authors cherry-picked the results that maximize the performances on the test, rather than choosing them on the validation set. The authors are strongly encouraged to honestly and openly comment on this; it is possible that my evaluation is incorrect, but the combination of lines 275-277 with 224-225, together with the caption of Table 4, seems to point in this direction.

Minor comments:

- while one usually assumes that the coarsening ratio reflects the percentage of retained nodes, it may be helpful to explicitly write this in the paper.
- It is known that a structure-agnostic baseline can produce state of the art results on PROTEINS, whereas it struggles on NCI1 for instance. It may have been a good idea to make an hyper-parameter study on the latter rather than the former dataset (but on the validation set)

---

> ### Author Response · Authors · 2022-07-29
> **Response to Reviewer k3VB**
>
> We thank the reviewer for the thoughtful review and for highlighting the quality of the paper. Before answering the raised questions, we would like to clarify some aspects of our evaluation procedure.
>
> ### Clarification of evaluation procedure
> 1 - As [6] is (to the best of our knowledge) the only paper tackling the size-generalization problem of GNNs in a graph classification setting with access only to small graphs during training, we followed exactly their evaluation procedure to properly compare against it. We used the same datasets, train/val/test splits, and hyperparameters (these were in fact obtained by the authors of [6] through an hyper-parameter tuning procedure to find the best configuration for the given datasets).
>
> 2 - To identify the values of $\lambda$ and $C$ to use for our method, we tried different values ($\lambda= \{1.0, 0.1, 0.01, 0.001\}$ and $C=\{ (0.5), (0.8), (0.9), (0.5,0.8), (0.5,0.9), (0.8,0.9)\}$) on the *validation* set using a GIN model on the PROTEINS dataset. We found that $\lambda=0.1$ and $C=(0.8,0.9)$ performed best on the *validation* set, and so we tried those on the validation set of other datasets and with other models. We noticed that results on the validation sets were good (as they were leading to better results on the validation sets with respect to a model trained without our regularization), and so we decided to keep the same values of $\lambda$ and $C$ for all datasets and models to show that our method can work without extensive (and expensive) hyperparameter tuning (and we believe this is further proof of the effectiveness of our method).
>
> 3 - We first obtained the results for Table 1, Figure 2, Table 2, and Table 3, in the main paper. Then, only *after* having those results, we have performed the ablation study. Please notice that, in the results shown in the ablation study, there are several settings which lead to higher results than what is shown in Tables 2 and 3. Using those results in Tables 2 and 3 would have made our method look even stronger, but clearly would have invalidated our evaluation procedure as we would have used the test set to find those configurations, which instead is not the case.
>
> To conclude, we remark that the results shown in the ablation study are __not__ coming from an hyperparameter tuning procedure, but were performed __after__ all the tuning (done on the validation set) and evaluation results were obtained. The ablation is used to understand the impact of the components of our method only *after* having evaluated it, as is the standard procedure for ablation studies. We have now specified this in the appendix, and we thank the reviewer for highlighting this aspect.
>
> ### Answer to questions
> - Q1. We thank the reviewer for highlighting this aspect. For space limitations we weren't able to include precise details for the procedure. We will however include these in the Appendix. The values of $\lambda$ used here is $0.1$, and the value of $C$ is $(0.8, 0.9)$ as for the results in Tables 2 and 3.
> - Q2. As the same coarsening ratio is applied to every graph in the dataset, all graphs are coarsened by the same amount. This means that (except in pathological cases with extremely small coarsening ratios), the "proportions" in the dataset will remain the same, in the sense that, for example, the ratio between the number of nodes in the largest graph and the number of nodes in the smallest graph should stay almost the same. With low coarsening ratios, there is however the problem that the coarsening can be too aggressive and the graphs may lose all their distinguishing topological information. This is one of the reasons why very low coarsening ratios lead to poor performance (as shown in our ablation study).
> - Q3. For our evaluation we followed the datasets proposed by the current state-of-the-art ([6]), and we analyzed the impact of different coarsening ratios in our ablation study. In this work we wanted to focus on introducing a method for improving size-generalization in GNNs, and we agree that trying our method on different kinds of networks and applications (and with application-specific design choices) is a great direction for future work.

---

> > ### Author Response · Authors · 2022-07-29
> > **Additional comments**
> >
> > Together with the previous response, we add some additional comments on some aspects raised be the reviewer.
> >
> > #### Other comments
> > - We thank the reviewer for highlighting that it is best to explicitly write that the coarsening ratio reflects the percentage of retained nodes.
> > - CKA is designed exactly for the purpose of comparing representations of neural networks, and is designed to work with representations of possibly different sizes. CKA is commonly used to study if different models are encoding the same information, or if different layers of the same model are containing the same information (see [32]).
> > - Regarding the computational complexity: generating the coarsened datasets is a *pre-processing* step that only needs to be done *once*, and here the complexity will depend on the choice of the coarsening algorithm. At training time, for a given batch, there is an additional forward pass for each considered coarsening ratio and then there is the computation of the loss. Both these operations require constant additional time, and in practice we notice an overhead of up to 50%, as written in the paper. At inference time there is no additional computation to be performed, and so no overhead of any kind.

---

> > > ### Author Response · Authors · 2022-08-05
> > > **Additional Experiment**
> > >
> > > We would like to add that we have performed an additional experiment as suggested by the reviewer.
> > >
> > > In fact, we have tested our method on a dataset of a different domain. In more detail we have tested it on a graph classification dataset of social networks: the Deezer dataset introduced in "Karate Club: An API Oriented Open-source Python Framework for Unsupervised Learning on Graphs", Rozemberczki et al., CIKM 2020.
> > >
> > > The results are shown below (results for a model trained with and without our regularization). We followed the same evaluation procedure as for Table 2 and Table 3 in the main paper: we train on the 50% smallest graphs and test on the 10% largest, and we employ $\lambda=0.1$ and $C=(0.8,0.9)$ without any sort of hyperparameter tuning. We notice our method proves highly effective with improvements of up to 20%.
> > >
> > > | Dataset | Deezer       | Deezer       |
> > > |---------|--------------|--------------|
> > > | Reg.    | No           | Yes          |
> > > | PNA     | 0.59 +- 0.06 | 0.64 +- 0.07 |
> > > | GCN     | 0.49 +- 0.10 | 0.59 +- 0.06 |
> > > | GIN     | 0.55 +- 0.08 | 0.61 +- 0.07 |
> > >
> > > We will include these results in the final version of the paper.

---

### Official Review · Reviewer_MVo5 · 2022-07-12

**Rating:** 5
**Confidence:** 4
**Soundness:** 2 fair
**Presentation:** 2 fair
**Contribution:** 3 good

**Summary:**

The paper proposes a new regularization strategy for improving size generalization in GNNs. The high-level idea consists of applying coarsening algorithms to original training data and then enforcing node embeddings from the original graph and coarsened ones to be closer. The paper considers four benchmarks for graph classification to assess the effectiveness of the proposal.



**Questions:**

- As stated in Section 3.1, the proposal assumes labels are not too sensitive to size. Is there any connection between this assumption and the chosen benchmarks? Why is the proposed method only validated using four molecular datasets?
- It would be helpful to have an ablation study considering different discrepancy metrics. For instance, would simple distance-based metrics obtain good results?
- The proposal seems to have the potential for improving pre-training. When evaluating existing strategies, people often split molecules according to their scaffold [1]. Showing gains to that task would increase the impact of the proposal.
- Accuracy results could go in the appendix.
- Showing training times would support the claim that the proposal incurs a 50% overhead.

[1] https://arxiv.org/pdf/1905.12265.pdf

**Limitations:**

The paper briefly describes the limitations of the proposal.

**Strengths And Weaknesses:**

Overall, the paper reads well and the proposed method is rather simple and general. Also, the problem of interest seems relevant.

However, the paper lacks stronger motivation behind some design choices, such as graph coarsening algorithms and discrepancy metrics. For instance, CMD is chosen because "it has proven to be successful and stable as a regularization term for non-linear models".  Also, the experimental setup is weak as it only includes four molecular datasets, and follows an existing setup (Bevilacqua et. al., 2021). Moreover, the analysis section provides no principled assessment of the method, mostly showing that the proposed regularization somehow affects node representations.

Strengths
- Simplicity: the idea is somehow intuitive even though it is not theoretically grounded.
- Promising results

Weaknesses
- Limited evaluation setup
- Poor understanding regarding the applicability of the proposal (only general comments are made in Section 3.1 --- Limitations)

---

> ### Author Response · Authors · 2022-07-29
> **Response to Reviewer MVo5**
>
> We thank the reviewer for the insightful comments, and for highlighting the importance of the problem and of our results.
> We answer below to the questions raised by the reviewer, and we remain available for further clarifications.
>
> ### Answer to questions
> - Q1. The choice of the datasets is simply coming from relevant prior work which we have to compare against. As [6] is (to the best of our knowledge) the only paper tackling the size-generalization problem of GNNs in a graph classification setting with access only to small graphs during training, we followed exactly their evaluation procedure to properly compare against it.
> - Q2. Our method is trying to compare *distributions* of node embeddings, so we needed distribution divergences, and not element-wise divergences. Considering the literature on the use of distribution-wise divergences in machine learning, the choice was between MMD and CMD. We chose the latter as MMD can be seen as a simpler version of CMD (in fact the two are exactly the same if one removes the higher order moments component from CMD), and previous works ([41,58,60]) highlighted the superiority of CMD (for example empirical results show that CMD is less susceptible to the weight with which the regularization is added to the loss).
> - Q3. We agree that there are many areas of interest in which our work can be applied, and we believe this is a sign of its potential impact and of the importance of the tackled problem. In this work we focused on proposing a new method for the problem of size-generalization that can be applied on any GNN. Given the space limitation of nine pages, we leave the specialised applications of our technique for future work.
> - Q4. As the datasets are strongly unbalanced (as can be seen from the dataset statistics in the appendix), the accuracy values provide no significant signal. Just to give an example, if you have a model that assigns *all graphs* in the test set to the majority class, it would achieve *at least* ~80% accuracy on *all* datasets, even though it would classify incorrectly *all* graphs belonging to the minority class. Using the MCC as a metric allows us to instead clearly study the quality of the models in this unbalanced setting. This is also confirmed in previous work ([6]) which only reports MCC values for the same reason.
> - Q5. We thank the reviewer for the remark, we have added the training times in the appendix to confirm the 50% overhead.

---

> > ### Author Response · Authors · 2022-08-05
> > **Additional Experiment**
> >
> > We would like to add that we have performed an additional experiment as suggested by the reviewer.
> >
> > In fact, we have tested our method on a dataset of a different domain. In more detail we have tested it on a graph classification dataset of social networks: the Deezer dataset introduced in "Karate Club: An API Oriented Open-source Python Framework for Unsupervised Learning on Graphs", Rozemberczki et al., CIKM 2020.
> >
> > The results are shown below (results for a model trained with and without our regularization). We followed the same evaluation procedure as for Table 2 and Table 3 in the main paper: we train on the 50% smallest graphs and test on the 10% largest, and we employ $\lambda=0.1$ and $C=(0.8,0.9)$ without any sort of hyperparameter tuning. We notice our method proves highly effective with improvements of up to 20%.
> >
> > | Dataset | Deezer       | Deezer       |
> > |---------|--------------|--------------|
> > | Reg.    | No           | Yes          |
> > | PNA     | 0.59 +- 0.06 | 0.64 +- 0.07 |
> > | GCN     | 0.49 +- 0.10 | 0.59 +- 0.06 |
> > | GIN     | 0.55 +- 0.08 | 0.61 +- 0.07 |
> >
> > We will include these results in the final version of the paper.

---

> > ### Comment · Reviewer_MVo5 · 2022-08-08
> > **Response to Authors**
> >
> > Thank you for your reply. I appreciate the effort to run additional experiments in a limited time.
> >
> > Overall, my concerns have been partially addressed. More details:
> >
> > Q1. [Partially addressed] While [6] considers the same four datasets, it also includes empirical analysis on unattributed graphs. Since the proposed method is not theoretically grounded, I believe it would require a stronger empirical assessment. The additional experiment on Deezer only partially addresses this issue.
> >
> > Q2. [Not addressed] I still think the authors could have considered, e.g., Wasserstein distance between the sets of node embeddings (original and coarsened graphs) or even applied l2-norm after readout layers.
> >
> > Q3. [Addressed] This was more a general comment than an important issue. I was not expecting authors to consider other setups given the limited time.
> >
> > Q4. [Addressed] My point was that reporting accuracy results would not hurt. But I have no concerns regarding MCC.
> >
> > Q5. [Addressed] It would be helpful to report the coarsening ratios associated with these experiments in the Appendix.

---

> > > ### Author Response · Authors · 2022-08-08
> > > **Further response to reviewer**
> > >
> > > We sincerely thank the reviewer for taking the time to answer our rebuttal. We will try to address the standing concerns below.
> > >
> > > - Q1. The unattributed datasets in [6] were designed to test the theoretical assumptions behind the model proposed in [6], and in fact there is a big discrepancy (in the ranking of the model score) with the results they then have on the "real-world" datasets. We then focused on the real world datasets to test the performance of our method in realistic benchmark datasets. The Embedding Analysis of Section 4 in the main paper was designed to provide insights on the intuition behind our method by studying how our method affects the node embeddings generated by the models. The results show in fact that models trained with our  strategy are more robust to size-shifts with respect to models trained without our strategy. CKA is a popular and powerful tool for analysing the representations of neural networks (see for example [Nguyen et al., ICLR 2021]).
> > >
> > > - Q2. Applying the L2 norm on the graph representations after readout layers would be a much weaker training signal than using a distribution-wise discrepancy metric such as CMD, which can be seen as performing multiple L2 comparison between different "readout representations" (one per every considered moment). In fact what is suggested by the reviewer could be seen as a special case of CMD in which we remove the second term (i.e. the summation over $k$). We can however incorporate some results in the Appendix of the camera ready version, and we thank the reviewer for the suggestion.

---

### Official Review · Reviewer_1c5i · 2022-07-12

**Rating:** 5
**Confidence:** 3
**Soundness:** 3 good
**Presentation:** 2 fair
**Contribution:** 2 fair

**Summary:**

The paper focuses on how to generalize GNNs to graphs of different sizes in graph classification. Previous works either add an ad-hoc strategy or require access to test graphs. In this work, the authors propose a regularization method based on graph coarsening techniques to improve the size-generalization of GNNs for graph classification. Empirical results show that the performance will be improved up to 30% with the proposed regularization method.

**Questions:**

The paper uses the SGC coarsening algorithm in the experiments. Do the authors try other coarsening methods and evaluate how the coarsening algorithm affects the regularization performance?

**Limitations:**

The authors discuss the limitation in Sec 3.1 about the assumption that there are some size-invariant properties that determine the label.

**Strengths And Weaknesses:**

Strengths:
* The paper proposes a regularization strategy that improves the size-generalization ability of GNNs for graph classification.
* Empirical results show that GNN models with the proposed regularization strategy achieve comparable or better size-generalization performance than baselines.

Weaknesses:
* The method is straightforward and the novelty is limited. The paper combines existing coarsening techniques and an existing metric (i.e. CMD) for the regularization.

---

> ### Author Response · Authors · 2022-07-29
> **Response to Reviewer 1c5i**
>
> We thank the reviewer for the time and comments. We believe that a novel combination of existing techniques to solve an important practical issue is a valuable contribution.
> We answer below to the questions raised by the reviewer, and we remain available for further clarifications.
>
> ## Answer to questions
> - Q1. Yes we do try different coarsening algorithms. The results are shown in the paper in Section 5.2 where there is a subsection titled "Changing Coarsening Method.", in particular in Figure 3 it's possible to notice how the performance change using 4 different coarsening methods. Two of these are specialised graph coarsening techniques, while the other two are baseline graph clustering methods. We notice that our regularization strategy is robust to the choice of coarsening algorithm.

---

### Author Response · Authors · 2022-08-05
**Additional Experiment**

We thank all the reviewers for their time and effort. Together with replying to all the questions, we have performed an additional experiment as suggested by reviewers MVo5 and k3VB.

In fact, we have tested our method on a dataset of a different domain. In more detail we have tested it on a graph classification dataset of social networks: the Deezer dataset introduced in "Karate Club: An API Oriented Open-source Python Framework for Unsupervised Learning on Graphs", Rozemberczki et al., CIKM 2020.

The results are shown below (results for a model trained with and without our regularization). We followed the same evaluation procedure as for Table 2 and Table 3 in the main paper: we train on the 50% smallest graphs and test on the 10% largest, and we employ $\lambda=0.1$ and $C=(0.8,0.9)$ without any sort of hyperparameter tuning. We notice our method proves highly effective with improvements of up to 20%.

| Dataset | Deezer       | Deezer       |
|---------|--------------|--------------|
| Reg.    | No           | Yes          |
| PNA     | 0.59 +- 0.06 | 0.64 +- 0.07 |
| GCN     | 0.49 +- 0.10 | 0.59 +- 0.06 |
| GIN     | 0.55 +- 0.08 | 0.61 +- 0.07 |

We will include these results in the final version of the paper.

We also kindly remind the reviewers that we remain available for further clarifications.

---

### Meta-Review · Area_Chair_WyW3 · 2022-09-02

**Recommendation:** Accept
**Confidence:** Certain

**Metareview:**

This work proposes a regularization approach (based on graph coarsening and alignment) to allowing graph neural networks to generalize across different graph sizes. The approach proposed here is simple, yet shown to be effective. While the reviewers had some concerns regarding this paper and the results in it, these were alleviated to a sufficient extent in rebuttal so that currently one reviewer outright supports acceptance, and the others lean towards acceptance. I agree with the opinion supporting acceptance of the paper, especially given the remark about the value (and rarity) of simple, clear, and effective solutions to important problems, which I agree is the case here. Therefore, I recommend accepting the paper, and I would like to encourage the authors to take into account the reviewers' comments (and the additional points included in the responses to them) when preparing the camera ready version.

**Award:**

No

---

### Decision · Program_Chairs · 2022-09-14

Accept